# Assessing the consequences of gestational diabetes mellitus on offspring's cardiovascular health: MySweetHeart Cohort study protocol, Switzerland

Stefano Di Bernardo,[1] Yvan Mivelaz,[1] Adina Mihaela Epure,[1,2] Yvan Vial,[3] Umberto Simeoni,[4] Pascal Bovet,[2] Sandrine Estoppey Younes,[2] Arnaud Chiolero,[2,5,6] Nicole Sekarski,[1] on behalf of MySweetHeart Research Group

SDB and YM contributed equally.

For numbered affiliations see end of article.

**Correspondence to**
Professor Nicole Sekarski;
nicole.sekarski@chuv.ch

## ABSTRACT

**Introduction** Gestational diabetes mellitus (GDM) is a state of glucose intolerance with onset during pregnancy. GDM carries prenatal and perinatal risks as well as long-term risks for the mother and her child. GDM may be involved in the foetal programming of long-term cardiovascular health. However, evidence is sparse and the effect of GDM on cardiovascular health is unknown. To address these issues, we will conduct MySweetHeart Cohort study. The objectives are to assess the effect of GDM on offspring's cardiovascular health early in life by using surrogate markers of cardiovascular disease and atherosclerosis.

**Methods and analysis** This is a cohort study of 100 offspring of women with GDM and 100 offspring of women without GDM. At inclusion, a baseline assessment of the mothers will be conducted through means of self-report questionnaires, a researcher-administrated interview, blood pressure and anthropometric measurements, and a maternal blood sampling. Between the 30th and 34th weeks of gestation, a foetal echography will be performed to assess the foetal cardiac structure and function, the fetomaternal circulation and the hepatic volume. At birth, maternal and neonatal characteristics will be assessed. An echocardiography will be performed to assess cardiac structure and function 2–7 days after birth; carotid intima-media thickness will be also measured to assess vascular structure. MySweetHeart Cohort is linked to MySweetHeart Trial (clinicaltrials.gov/ct2/show/NCT02890693), a randomised controlled trial assessing the effect of a multidimensional interdisciplinary lifestyle and psychosocial intervention to improve the cardiometabolic and mental health of women with GDM and their offspring. A long-term follow-up of children is planned.

**Ethics and dissemination** Ethical approval has been obtained through the state Human Research Ethics Committee of the Canton de Vaud (study number 2016–00745). We aim to disseminate the findings through regional, national and international conferences and through peer-reviewed journals.

**Trial registration number** ClinicalTrials.gov (clinicaltrials.gov/ct2/show/NCT02872974).

## Strengths and limitations of this study

► This study will increase our understanding of the impact of gestational diabetes mellitus on cardiometabolic health during foetal and neonatal periods. Within a Developmental Origin of Health and Disease framework, it will contribute to improving the knowledge on the early life conditioning (programming) mechanisms and their influence on infants' cardiovascular health.

► A major strength of our study is the assessment at birth of surrogate markers of cardiovascular disease (CVD) (left ventricular mass index and subclinical atherosclerosis (carotid intima-media thickness)).

► Acknowledged limitations are the relatively small sample size, the potential for residual confounding and the limited representativeness of our sample (monocentric study).

► A better understanding of the clinical impact of the maternal hyperglycaemic disorders on offspring's health early in life is needed to inform strategies for the early prevention of CVD.

► A long-term follow-up of children is planned for further assessment of their cardiovascular health.

## INTRODUCTION
### Maternal hyperglycaemic disorders during pregnancy

The prevalence of maternal hyperglycaemic disorders during pregnancy is increasing in the population due to the obesity epidemic and the increasing age of pregnant mothers.[1–3] Maternal hyperglycaemic disorders include type 1 and 2 diabetes mellitus (T1/T2DM) and gestational diabetes mellitus (GDM). T1/T2DM is a state of glucose intolerance occurring independently of pregnancy, while GDM is characterised by glucose intolerance that begins or is first diagnosed during pregnancy and usually resolves after delivery.[4 5] GDM is

associated with an increased risk of maternal metabolic syndrome and T2DM[2 3 6] and is now the most frequent form of hyperglycaemic disorders during pregnancy.[2]

Maternal T1/T2DM diabetes is associated with significant fetal and neonatal morbidity,[4] for example, large for gestational age (LGA), congenital malformations and neonatal metabolic disorder (hypoglycaemia).[3 7 8] There is some evidence that the foetal consequences are more serious with T1/T2DM diabetes than with GDM[4] and, until recently, the consequences of GDM on offspring's health were uncertain.[9] Nevertheless, the large high-sensitiviy C reactive protein (HAPO) cohort study showed that maternal hyperglycaemia less severe than in T1/T2DM, after correction for potential confounders (maternal obesity, blood pressure), was also associated with adverse perinatal outcomes, that is, LGA, neonatal hyperinsulinism and hypoglycaemia, and pre-eclampsia.[9–11] Furthermore, in large randomised trials, treatment of GDM resulted in better perinatal outcomes, including a reduced risk of LGA.[4 12–15]

Following the results of these studies, the International Association of Diabetes and Pregnancy Study Groups (IADPSG) have proposed new criteria to diagnose GDM.[3] The benefits of screening and treatment of GDM have long been debated[14–17] and recommendations from medical societies varied from no screening at all, screening only high-risk women (risk-based screening), to universal screening. Thus, the IADPSG recommend universal one-stage screening procedure that includes a 2-hour 75 g oral glucose tolerance test (OGTT). A single abnormal glucose concentration at fasting, at 1 hour or at 2 hours is sufficient for the diagnosis of GDM.[14] New blood glucose threshold were derived from the HAPO study and based on pregnancy outcomes associated with the glucose level.[10 14] Using these criteria, the prevalence of GDM is expected to be high, reaching up to 10%–20% of pregnant women in US populations.[3]

## CARDIOMETABOLIC EFFECTS OF MATERNAL DM

Maternal hyperglycaemic disorders are associated with fetal cardiac and vascular structural and functional alterations. For instance, it is well known that offspring of T1/T2DM mothers are at increased risk of congenital cardiac malformations and cardiac hypertrophy.[8 18] In a retrospective study of 92 offspring of 87 diabetic mothers conducted in Lausanne, we observed that 5 neonates had congenital heart disease and 12 had ventricular hypertrophy.[8] There is also growing evidence that maternal hyperglycaemic disorders may be involved in the 'foetal programming' of obesity and metabolic disorders in their offspring.[11 19] Within a Developmental Origin of Health and Disease (DOHaD) framework,[20 21] fetal programming (or conditioning) is the process involved in the associations between the exposure to detrimental factors during foetal life and health outcomes later in life.[22] Foetal programming is suspected to be involved in the development of cardiometabolic disorders, such as elevated blood pressure, coronary heart diseases or DM, notably through epigenetic mechanisms.[20 21] Despite recent and large growth in this research area, studies are needed to better objectively characterise both early life exposures and cardiometabolic outcomes.[23]

Intrauterine exposure to high maternal blood glucose is associated with foetal hyperinsulinism which is responsible for structural and functional changes affecting mostly the liver and the cardiovascular system. The liver is the major metabolic organ, considered as the metabolic brain and responsible for the distribution of the placental resources.[24] Recent studies have shown that macrosomic offspring had impaired liver blood perfusion that contributes to foetal growth alteration[25–27] and that differential perfusion of the foetal liver modified hepatic metabolic function and size.[24] Studies indicate that the liver is enlarged in foetuses of T1/T2 DM mothers.[28 29] Furthermore, and similarly the foetal heart response to hyperinsulinism is the development of an asymmetrical hypertrophy predominant at the septal wall. The evaluation of liver volume, interventricular septum thickness and left ventricular mass is therefore of outmost importance and may represent surrogate markers of the foetal metabolic response to maternal GDM.

## LACK OF DATA ON THE EFFECTS OF GDM ON OFFSPRING'S CARDIOVASCULAR HEALTH

Most studies on the effect of maternal hyperglycaemic disorders were conducted among mothers with T1/T2DM, not among mothers with GDM and with an emphasis on the effects on offspring's metabolic outcomes rather than on offspring's cardiovascular health. It is also unknown whether treatment of GDM is effective to reduce long-term risk of offspring's metabolic and cardiovascular disease (CVD) risk[9 11 15] and the new definition of GDM calls also for the conduction of further studies on the impact of GDM on offspring's cardiovascular health. To study the early development of atherosclerosis and pathogenesis of CVD, it has become central to assess surrogate markers of CVD such as increased cardiac mass (left ventricular mass index; LVMI)[30] and of subclinical atherosclerosis (assessed through carotid intima-media thickness; cIMT).[30 31] These markers have been used extensively in studies in children and young adults with CVD risk factors.[30–34] However, to our knowledge, the effects of GDM on offspring's foetal and early neonatal cardiovascular health, and in particular on such surrogate markers,[31] have never been studied.

## OBJECTIVES

Our objectives are to assess the effect of GDM on offspring's cardiovascular health early in life, namely on (1) the surrogate markers of CVD and atherosclerosis at birth (LVMI and cIMT) (primary outcomes) and (2) the cardiovascular structure and function during the foetal

period and potential neonatal cardiovascular risk factors (secondary outcomes).

## METHODS AND ANALYSIS
### Study design
We plan to conduct a cohort study of 200 offspring of women aged 18 years old or more. We will include 100 women with GDM and 100 without GDM. All pregnant women attending the antenatal care clinic or the GDM clinic at the Lausanne University Hospital (CHUV) will be invited to participate. In addition, women with GDM who are followed up by private diabetologists and gynaecologists in the Canton de Vaud will be invited to participate following a similar procedure. Using the new criteria of the IADPSG, the expected prevalence of GDM in this population is at least ~10%.[3] Women with T1/T2D will not be included.

## STUDY POPULATION
Inclusion criteria: Pregnant women aged 18 years or older, with or without GDM at 24–32 weeks of gestation, and understanding French or English. Exclusion criteria: Women on strict bed rest, with pre-existing DM or severe mental disorders precluding participation.

## STUDY OUTCOMES
Primary outcomes are surrogate markers of CVD and atherosclerosis at birth, that is, LVMI and cIMT. Secondary outcomes are foetal cardiovascular structure and function (notably LVMI and liver volume) and potential neonatal cardiovascular risk factors (including cord blood lipid levels, glycaemia, insulin, high-sensitiviy C reactive protein (hs-CRP), etc).

## DATA COLLECTION AND VISITS
During pregnancy, in the pre-recruitment phase, all pregnant women without pre-existing DM, attending the CHUV antenatal care clinic, have a fasting blood glucose measurement between the 24th and 28th weeks of gestation. GDM is diagnosed if fasting blood glucose is equal to or above 5.1 mmol/L. If fasting blood glucose is equal to or above 4.4 and below 5.1 mmol/L, pregnant women have a 2-hour 75 g OGTT and GDM will be diagnosed based on the new criteria of the IADPSG.[3 9] Women screened by private obstetricians in the Canton de Vaud undergo either the same procedure or directly a 2-hour 75 g OGTT. Considering the variability in patients' compliance and time of enrolment for prenatal care, all eligible women, having been screened for GDM before 32 weeks of gestation, will be invited to participate (see inclusion and exclusion criteria in the Study population section).

The participation in the study is voluntary and involves a sequence of events and measurements as summarised in table 1.

At the first study visit (V1), scheduled between the 24th and 32nd weeks of gestation, a baseline assessment of the mothers will be conducted through means of self-report questionnaires and a researcher-administered interview, weight, height and blood pressure measurements, and a maternal blood sampling. This will provide information notably on the socioeconomic status, general health status, family medical history of CVD or diabetes, smoking and drinking habits and several cardiometabolic biomarkers, all of which will be treated as potential confounding factors. If available, paternal data on the family history of CVD or diabetes will be also collected at the first study visit.

At the second visit (V2), between the 30th and 34th weeks of gestation, a foetal echography will be performed to assess the fetal cardiac structure (including LVMI) and function, the fetomaternal circulation, and the hepatic volume.

In women with GDM, information about treatment and blood glucose measurements will be recorded during the whole pregnancy.

At birth (V3), maternal characteristics (including mode of delivery, gravidity, parity, etc) and neonatal characteristics (including gestational age, sex, weight, length, etc) will be assessed. A sample of umbilical cord blood will be collected to measure various biomarkers, including glucose, insulin, total cholesterol, low-density lipoprotein (LDL)-cholesterol, high-density lipoprotein (HDL)-cholesterol, triglycerides, uric acid, creatinine, hs-CRP and various miRNA.

An echocardiography will be done to assess cardiac structure (including LVMI) and function 2–7 days after birth (V4). cIMT will be measured to assess vascular structure.

## MEASUREMENTS METHODS
### Maternal clinical assessment
Maternal body weight and blood pressure (three readings at 2-min intervals with a clinically validated[35] oscillometric sphygmomanometer (OMRON HEM-907)[36 37]) will be measured during pregnancy (V1, V2) and in postpartum (V4). Clinical chemistry (including glucose, hemoglobinA1c (HbA1c), insulin, total cholesterol, LDL-cholesterol, HDL-cholesterol, triglycerides, uric acid and creatinine) will be assessed at the first study visit (V1) and blood will be collected for further potential analyses. All samples will be managed and stored in a biobank. The clinical data assessed as part of the usual care of the mother and potentially influencing the outcomes of interest will be collected. Also, information on socioeconomic characteristics, general health, family history of CVD or diabetes, smoking and drinking habits and so on will be gathered using a self-report questionnaire and a researcher-administered interview.

**Table 1** Schedule of events and main assessments

| | Visit | Period (WG) | Mother | Fetus/newborn |
|---|---|---|---|---|
| Pregnancy | V1 (Baseline) | 24th to 32nd | ▲ Information notably on socioeconomic characteristics, general health, family history of CVD or diabetes and smoking and drinking habits<br>▲ Weight and height measurements<br>▲ Three blood pressure measurements<br>▲ Blood sampling | ▲ N/A |
| | V2 | 30th to 34th | ▲ Weight<br>▲ Three blood pressure measurements | ▲ Foetal cardiac echography |
| Delivery or postdelivery | V3 | Birth | ▲ Blood sampling | ▲ Cord blood sampling, newborn weight, length and other birth-related information |
| | V4 | 2–7 days after birth | ▲ Weight<br>▲ Three blood pressure measurements | ▲ Neonatal cardiac echography and carotid echography |

CVD, cardiovascular diseases; N/A, not applicable; WG, weeks of gestation.

## Offspring clinical assessment

The height and weight of the neonate will be measured and the growth reference charts of the Swiss Society of Paediatrics will be used.[38 39] Gestational age will be determined using either the date of the last menstrual period or the first-trimester ultrasound dating. Clinical chemistry (including glucose, total cholesterol, LDL-cholesterol, HDL-cholesterol, triglycerides, insulin, uric acid, hs-CRP, mi-RNA) will be assessed at birth using cord blood. We will explore the profile of circulating non-coding RNAs in offspring of women with GDM and without GDM to assess the involvement of epigenetic mechanisms in cardiovascular alterations.[40] Blood for future analyses will be stored in a biobank. The clinical data assessed as part of the usual care of the offspring and potentially influencing the outcomes of interest will be collected.

## Foetal echography

oThis examination will be performed between the 30th and 34th weeks of gestation by two experienced ultrasonographers (YM, YV), blinded to the maternal glycaemic status, using a Voluson E10 (General Electric Medical Systems, ZIPF, Australia) equipped with a 2–8 MHz convex array sector transducer.[41] A systematic echography will be conducted to detect structural and functional anomalies following international recommendations.[42 43] Anthropometric measurements and foetal well-being parameters will be measured.[44] Observed foetal weight (FW) will be calculated using the Hadlock *et al*'s formula based on the head circumference, biparietal diameter, abdominal circumference and femur length.[45] The gestational age-adjusted expected FW will be calculated[46] and the discordance with observed FW will be determined as (observed FW−expected FW)/expected FW. This discordance will be used to quantify appropriateness of foetal growth. Then, a detailed echocardiography of the foetal liver, cardiovascular system and of the maternofoetal circulation will be conducted including bidimensional (2D), M-Mode (M), Doppler (D) and tridimensional (3D) imaging of the following structures: cardiac four chambers (2D), right ventricular (RV) and left ventricular (LV) short axis (2D, M), tricuspid, pulmonary, mitral and aortic valve (2D, D), umbilical vein, ductus venosus, patent ductus arteriosus, aortic isthmus, umbilical arteries and maternal uterine arteries (2D, D) and liver (2D, 3D).[47–50] This will allow the assessment of the LV posterior wall and septal thickness, the LV mass,[51] the LV and RV systolic and diastolic function, the cardiothoracic ratio, the placental and foetal vascular resistances, the cardiac output and liver volume. Data will be digitally stored to allow off-line analyses. Measures will be expressed as Z-scores based on gestational age to allow comparison between foetuses.

## Neonatal cardiac echography

This examination will be performed 2–7 days after birth (before the mother and the newborn leave the clinic) by two experienced ultrasonographers (NS, SDB) blinded

to the maternal glycaemic status and to the foetal echocardiogram. Echocardiography will be performed on a Philips EPIC echocardiogram with a S8-3 or S12 MHz transducer, digitally recorded (Xcelera, Philips) and analysed off-line. Standard echocardiography including 2D, M-mode, colour and spectral Doppler according to the American Society of Echocardiograph (ASE) will be performed.[52 53] Measurements will be performed according to the standard of the ASE[52 53] including: (1) right and left atrial chamber sizes; (2) LV size, volume and function; (3) LV mass; (4) LV and RV systolic and diastolic function. Z-scores based on BSA calculated by the Haycock formula will be used to express these measurements.[54–56]

### Neonatal carotid echography

This examination will be performed 2–7 days after birth (before the mother and the newborn leave the clinic) by two experimented ultrasonographers (NS, SDB) blinded to the maternal glycaemic status. Carotid IMT measurement will be performed on a Philips EPIC echocardiograph (Philips Medical, Netherland) with a L 11–5 MHz high-resolution linear array transducer, recorded on a digital system (QLab, Philips Medical Netherlands). Image acquisition will be done according to the standard of the American Heart Association Atherosclerosis, Hypertension and Obesity in Youth Committee of the Council on CVD in the Young.[31] Image analyses will include: (1) far-wall cIMT; (2) calculation of the mean of maximal cIMT measurements; (3) calculation of carotid stiffness. Measurements of cIMT will be expressed in millimetre as mean±SD and compared with normal cIMTs.[57 58] We have recently conducted a study to show that the measurement of cIMT was feasible in young non-sedated infants. This study conducted among 81 infants less than 1 year of age confirmed that cIMT was measurable with a high interobserver reliability (coefficient of variation: 5.9%).[59]

### ANTICIPATED RECRUITMENT

Recruitment has begun in September 2016. About 3200 deliveries per year occur at our hospital and about 250 pregnant women are followed for GDM at our specialised clinic. A relatively large share of patients consists of immigrants who do not speak English or French and will not be eligible for participation. We therefore expect to include up to two pregnant women per week, hence to include the 200 subjects in 30 months. A research coordinator (SE) and a PhD student (AME) will be responsible to screen all women for eligibility and to ensure follow-up throughout the study duration. Recruitment will be facilitated by the high support of obstetricians at the CHUV. We will also benefit of the new specialised GDM clinic of the CHUV where every pregnant woman who screened positive for GDM at the CHUV antenatal care clinic is followed up. MySweetHeart Cohort is linked to MySweetHeart Trial (clinicaltrials.gov: NCT02890693; led by Professor Jardena Puder and PD Dr Antje Horsch), a randomised controlled trial led by this new specialised GDM clinic designed to assess the effect of a multidimensional interdisciplinary lifestyle and psychosocial intervention to improve the cardiometabolic and mental health of women with GDM and their offspring. Little loss to follow-up is expected because measurements are made during pregnancy and shortly after delivery. Nevertheless, to minimise loss to follow-up, most appointments are made on the days of regular follow-up visits and the postnatal echocardiography is scheduled before the mother and the baby leave the hospital.

### SAMPLE SIZE

Assuming that LVMI will be $30\,\text{g/m}^{2.7}$ (SD: 4.5) in children of mothers without GDM[60] compared with $32\,\text{g/m}^{2.7}$ (SD: 4.5) in children of women with GDM, the required sample

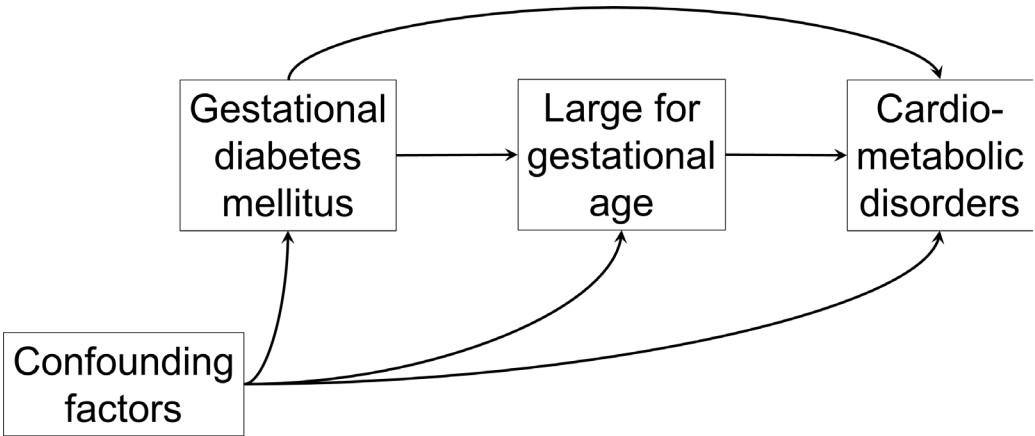

**Figure 1** Hypothetical and highly simplified causal relationships between gestational diabetes mellitus (GDM) and cardiometabolic disorders. Two pathways can be hypothesised: (1) a direct effect of GDM on offspring's cardiometabolic disorders; (2) an indirect effect through the mediator large for gestational age (LGA). It means that if LGA was prevented, part of the effect of GDM on cardiometabolic disorders would be prevented. Several confounding factors (such as maternal obesity, weight gain during pregnancy, smoking, socioeconomic status or family history of cardiometabolic diseases) have to be accounted.

size is 80 women with GDM and 80 women without GDM to have a statistical significant difference with a power of 80% and an alpha-level set at 0.05 (two sided). This sample size is also sufficient to observe statistical significant differences in cIMT at birth if we assume that cIMT at birth is 0.44 mm (SD: 0.04)[31] in newborn of women with GDM and 0.42 mm (SD: 0.04) in newborn of women without GDM. In the worst case scenario, we could lose up to 20% of participants between inclusion and delivery. Therefore, we plan to include 100 pregnant women with GDM and 100 women without GDM.

## STATISTICAL ANALYSIS

The association between exposure (GDM/no GDM) and the (continuous) outcomes will be estimated with linear regression analyses. Adjustment for potential confounding factors (maternal characteristics such as age, body mass index, blood pressure, socioeconomic status, personal history of CVD, treatment for GDM during pregnancy) will be done. To assess the direct and indirect (through mediators) effects of GDM on the outcomes, causal mediation analyses will be conducted.[61 62] Because LGA is a potential mediator of the effect of GDM on offspring outcomes,[63] analyses adjusted on birth weight will be conducted to estimate the indirect effect of GDM on the outcomes (figure 1). We will also consider the effect of treatment which will be treated as a mediator between GDM, LGA and outcomes.

## ETHICS AND DISSEMINATION

The study poses little to no risk to participants and their offspring. Signed informed consent is obtained from all participating women. Participation in the study does not interfere with the typical care patients receive during pregnancy and after delivery. Clinicians in the obstetrical clinic provide clinical follow-up and, if necessary, participants are referred on for additional clinical management.

Results from this study will be disseminated at regional and international conferences and in peer-reviewed journals.

## SIGNIFICANCE AND OUTLOOK

The current high prevalence of maternal obesity and associated hyperglycaemic disorders may lead to a perpetuating intergenerational cycle of increasing obesity, metabolic and cardiovascular disorders. Well-designed prospective studies are necessary to better quantify the actual impact of maternal hyperglycaemic disorders on offspring's health and to identify the potential underlying mechanisms. Despite recent significant advances in DOHaD and foetal programming research, major challenges have to be addressed, notably by better objectively characterising both early life exposure and cardiometabolic outcomes.[23]

Indeed, many past studies on the foetal programming of cardiometabolic disorders suffered from some methodological flaws including confounding, no or poor standardisation of exposure and outcome assessment, and selection bias. Our prospective population-based study in pregnant women whose offspring is examined during fetal life and at birth avoids some of these issues. The long period needed for CVD to develop could be a limitation, but this will be overcome in our study by measuring surrogate outcomes (including LVMI and cIMT) that are well-established precursors of CVD.[31] In addition, a long-term follow-up of the offspring is planned (not funded yet) for further assessment of their cardiovascular health.

By looking at the fetomaternal circulation, foetal cardiovascular system and liver and foetal growth and CVD risk markers at birth, we will acquire unique data to gain insight in the early mechanisms underlying the development of CVD and metabolic diseases and on the role of maternal hyperglycaemic disorders. GDM is an opportunity for timely intervention for the prevention of cardiometabolic disorders in mothers and their offspring.[9] With the new criteria to define GDM, a growing proportion of women will be concerned by this disorder. Having a better understanding of the clinical impact of maternal hyperglycaemic disorders on offspring's health early in life is urgently needed to inform strategies for the early prevention of CVD.

**Author affiliations**
[1]Paediatric Cardiology Unit, Woman-Mother-Child Department, Lausanne University Hospital, Lausanne, Switzerland
[2]Institute of Social and Preventive Medicine (IUMSP), Lausanne University Hospital, Lausanne, Switzerland
[3]Obstetrics and Gynaecology Division, Woman-Mother-Child Department, Lausanne University Hospital, Lausanne, Switzerland
[4]DOHaD Laboratory, Paediatrics Division, Woman-Mother-Child Department, Lausanne University Hospital, Lausanne, Switzerland
[5]Department of Epidemiology, Biostatistics and Occupational Health, McGill University, Montreal, Canada
[6]Institute of Primary Health Care (BIHAM), University of Bern, Bern, Switzerland

**Collaborators** The following are members of MySweetHeart research group (including the authors of the present article), listed in alphabetical order: Pascal Bovet, MD MPH (Institute of Social and Preventive Medicine (IUMSP), Professor, Lausanne University Hospital, Lausanne, Switzerland; pascal.bovet@chuv.ch). Arnaud Chiolero, MD PhD, Senior lecturer (Institute of Social and Preventive Medicine (IUMSP), Lausanne University Hospital, Lausanne, Switzerland; Department of Epidemiology, Biostatistics and Occupational Health, McGill University, Montreal Canada; Institute of Primary Health Care (BIHAM), University of Bern, Switzerland; arnaud.chiolero@chuv.ch), Stefano Di Bernardo, MD, Lecturer, (Pediatric Cardiology Unit, Woman-Mother-Child Department, Lausanne University Hospital, Lausanne, Switzerland; stefano.di-bernardo@chuv.ch), Adina Mihaela Epure, MD (Pediatric Cardiology Unit, Woman-Mother-Child Department; Institute of Social and Preventive Medicine (IUMSP), Lausanne University Hospital, Lausanne, Switzerland; adina-mihaela.epure@chuv.ch), Sandrine Estoppey Younes, Msc (Institute of Social and Preventive Medicine (IUMSP), Lausanne University Hospital, Lausanne, Switzerland; sandrine.estoppey@chuv.ch), Leah Gilbert, MSc (Division of Endocrinology, Diabetes and Metabolism, Department of Medicine, Lausanne University Hospital, Lausanne, Switzerland; leah.gilbert@chuv.ch), Justine Gross, HES (Division of Endocrinology, Diabetes and Metabolism, Department of Medicine, Lausanne University Hospital, Lausanne, Switzerland; justine.gross@chuv.ch), Antje Horsch, DClinPsych, assistant Professor, Woman-Mother-Child Department, Lausanne University Hospital, Lausanne, Switzerland; antje.horsch@chuv.ch), Bengt Kayser, PhD, Professor (Institute of Sport Sciences, University of Lausanne; bengt.kayser@unil.ch), Stefano Lanzi, PhD (Division of Endocrinology, Diabetes

and Metabolism, Department of Medicine, Lausanne University Hospital, Lausanne, Switzerland; stefano.lanzi@chuv.ch), Yvan Mivelaz, MD, Lecturer (Pediatric Cardiology Unit, Woman-Mother-Child Department, Lausanne University Hospital, Lausanne, Switzerland; yvan.mivelaz@chuv.ch), Jardena J. Puder, MD, Professor (Division of Endocrinology, Diabetes and Metabolism, Department of Medicine, Lausanne University Hospital, Lausanne, Switzerland; jardena.puder@chuv.ch), Nicole Sekarski, MD, Professor (Pediatric Cardiology Unit, Woman-Mother-Child Department, Lausanne University Hospital, Lausanne, Switzerland; nicole.sekarski@ chuv.ch), Umberto Simeoni, MD, Professor (DOHad Laboratory, Pediatrics Division, Woman-Mother-Child Department, Lausanne University Hospital, Lausanne, Switzerland; umberto.simeoni@chuv.ch), Yvan Vial, MD, Professor (Obstetrics and Gynecology Division, Woman-Mother-Child Department, Lausanne University Hospital, Lausanne, Switzerland; yvan.vial@chuv.ch).

**Contributors** NS, AC, SDB and YM designed the study with input from all other authors. SDB and YM drafted the manuscript and contributed equally to the present work. US, PB, AME, SE, YV and AC significantly contributed to the establishment and refinement of study procedures and critically revised the manuscript. All authors approved the final version of the manuscript.

**Funding** This study is funded by the Swiss National Science Foundation (Project 32003B-163240). The study protocol has been peer-reviewed by national and international experts before being funded.

**Competing interests** None declared.

**Ethics approval** The study has been approved by the state Human Research Ethics Committee of the Canton deVaud (study number 2016-00745).

**Provenance and peer review** Not commissioned; externally peer reviewed.

**Data sharing statement** It is the intention of the authors that once MySweetHeart Cohort is completed and the research articles emerging from the study have been published, the data set supporting the results and conclusions presented in the respective research articles to be made available in the DATA@ IUMSP repository, .

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
