## [Reviewer comments · BMJ Open]

ARTICLE DETAILS

TITLE (PROVISIONAL)	Assessing the consequences of gestational diabetes mellitus on offspring's cardiovascular health: MySweetHeart Cohort study protocol, Switzerland.
AUTHORS	Di Bernardo, Stefano; Mivelaz, Yvan; Epure, Adina Mihaela; Vial, Yvan; Simeoni, Umberto; Bovet, Pascal; Estoppey, Sandrine; Chiolero, Arnaud; Sekarski, Nicole

VERSION 1 – REVIEW

REVIEWER	COSSON AP-HP, Bondy France
REVIEW RETURNED	03-May-2017

GENERAL COMMENTS	Reviewing a presentation paper is not as reviewing a research paper: everything is already written, accepted, founded... Thank you for this work to be done, which is well-presented
---

REVIEWER	Kelli Ryckman University of Iowa, USA
REVIEW RETURNED	16-Jun-2017

GENERAL COMMENTS	The authors present a protocol for the MySweetHeart Cohort study to examine the effect of GDM on offspring's cardiovascular health in the first week of life. The following are suggestions for improving clarity of the protocol: 1) In the abstract and throughout the manuscript the authors state the objective as assessing the effect of GDM on offspring's cardiovascular health early in life. However, the protocol actually is presenting assessment within the first week of life. While it is noted that the intent is to follow children longitudinally there is very little to no specific details on how this will be done. Therefore, it is recommended that the objective reflect this protocol and refer to the cardiovascular health of the offspring within the first week of life, not "early in life".2) Please discuss how the coordinators will prevent loss to follow-up and procedures for follow-up like reminder phone calls, etc. Please also address how withdraws will be handled.3) Please provide how many births occur annually at the hospital
---

	and what percentage are GDM to give a better sense for how recruitment will proceed. Also it states all women who meet inclusion criteria will be invited to participate but then that approximately 2 will be enrolled a week. I assume more than 2 will be eligible a week at this institution. How will that be handled? 4) Because of the limited sample size adjustment statistically for all of the very important confounders may not be adequate. I recommend the authors consider matching case to control subjects by at least BMI and parity if not also age.
--	--

REVIEWER	Delphine Mitanchez APHP, Université Pierre et Marie Curie, Paris VI Pôle de périnatalité, Service de néonatalogie Hôpital Armand Trousseau 26 avenue du Dr Arnold Netter 75012 Paris, France
REVIEW RETURNED	16-Jun-2017

GENERAL COMMENTS	This paper presents a study protocol. This study aims to evaluate cardiovascular risk factors in a cohort of newborn of mothers with gestational diabetes compared to newborns of mothers without diabetes. The evaluation is based primarily on cardiac and vascular ultrasound data at birth. The theme of programming is an important topic. There is little data on the specific subject of cardiovascular risk factors identified from the perinatal period. The protocol is well described but it needs minor revisions. The introduction of the paper is too long. Two paragraphs could be deleted p 5, text between “Several epidemiological...prepubertal offspring”. In the method part, what the authors call neonatal cardiovascular risk factors (secondary outcomes) are in fact biological factors (metabolic and inflammatory). There is currently no evidence of an association between the values of these factors in cord blood and cardiovascular risk factors. Offspring assessment: Could the authors give precision about miRNA they aim to study in cord blood. Moreover, details should be provided on the growth parameters collected at birth and the reference curves used. Fetal echocardiography: which reference curves will be used to determine the z-score of the cardiac ultrasound measurements. The authors mention in the statistical analysis the consideration of maternal treatment for diabetes. How will they take into account the quality of maternal diabetes control that can influence fetal development?
--

REVIEWER	Tine Dalsgaard Clausen Department of Gynecology and Obstetrics Nordsjællands Hospital, Hillerød 3400 Hillerød Denmark
REVIEW RETURNED	17-Jun-2017

GENERAL COMMENTS	This is a protocol describing a study, which started inclusion in September 2016. It is of course an editorial decision to publish research- protocols, however I would personally only very seldom be a reader of protocols as presented here. The scope to evaluate cardiovascular risk in GDM offspring is very relevant. It is measuring findings from fetal echo as well as echo in newborn+carotis intima measures as proxy for later cardiovascular risk in offspring of mothers with GDM compared to non-GDM mothers. The authors refers to a single reference claiming that the methods (when used in children and adolescents) are valid predictors of cardio-vascular disease in later life - However I am not convinced that the methods have been validated when used in fetuses and newborn. A long-term follow up would therefore be extremely important to evaluate whether early changes is associated with measures later in childhood – newborn cardiomyopathy is often reversible. However – a long-term follow up from 2x100 offspring will probably be underpowered due to lost-to follow-up, and it has not been funded yet. Furthermore the authors claim that the study has several advantages compared to other follow-up studies –I agree that the prospective design is admirable, however it does not at all eradicate risk of confounding, which the authors claims. Furthermore inconsistent results from previous studies cannot be referred to as “methodological flaws”. I miss data on intrauterine growth (UL sound measurers). Finally, as I read the protocol it is linked to another protocol – a RCT which aim to evaluate the effect of intensified intervention in GDM pregnancies. This is a relevant scope; however inclusion of 100 GDM pregnancies would be extremely underpowered to evaluate any effect . So - the study will in theory be able to evaluate early cardio-vascular risk indicators, however it will not have data to evaluate potential effects of a more tight glycemic on the cardiovascular risk in the offspring.
--

VERSION 1 – AUTHOR RESPONSE

Reviewer: 1

Reviewer Name: E Cosson

Reviewing a presentation paper is not as reviewing a research paper: everything is already written, accepted, founded.... Thank you for this work to be done, which is well-presented

Response: Thanks for this comment. As raised by the reviewers, this protocol had already been peer-reviewed and amended accordingly.

Reviewer: 2

Reviewer Name: Kelli Ryckman

The authors present a protocol for the MySweetHeart Cohort study to examine the effect of GDM on offspring's cardiovascular health in the first week of life. The following are suggestions for improving clarity of the protocol:

1) In the abstract and throughout the manuscript the authors state the objective as assessing the effect of GDM on offspring's cardiovascular health early in life. However, the protocol actually is presenting assessment within the first week of life. While it is noted that the intent is to follow children longitudinally there is very little to no specific details on how this will be done. Therefore, it is recommended that the objective reflect this protocol and refer to the cardiovascular health of the offspring within the first week of life, not "early in life".

Response: We agree that "early in life" is not specific enough. We will collect data during foetal life and after birth but we are not convinced that "within the first week of life" would be more satisfactory. We have therefore kept the term "early in life" but we were cautious to specify systematically that we collected data during foetal life and after birth.

2) Please discuss how the coordinators will prevent loss to follow-up and procedures for follow-up like reminder phone calls, etc. Please also address how withdraws will be handled.

Response: We have added the following sentence in the method section: "Little loss to follow-up is expected because measurements are made during pregnancy and shortly after delivery. Nevertheless, to minimise the loss to follow-up, most appointments are made on the days of regular follow-up visits and the postnatal echocardiography is scheduled before the mother and the baby leave the hospital."

3) Please provide how many births occur annually at the hospital and what percentage are GDM to give a better sense for how recruitment will proceed. Also it states all women who meet inclusion criteria will be invited to participate but then that approximately 2 will be enrolled a week. I assume more than 2 will be eligible a week at this institution. How will that be handled?

Response: We have added the following information in the methods section: "About 3200 deliveries per year occur at our hospital and about 250 pregnant women per year are followed for GDM at our specialised clinic. A relatively large share of patients consists of immigrants who do not speak English or French and will not be eligible for study participation. We therefore expect to include up to 2 pregnant women per week, hence to include the 200 subjects in 30 months."

4) Because of the limited sample size adjustment statistically for all of the very important confounders may not be adequate. I recommend the authors consider matching case to control subjects by at least BMI and parity if not also age.

Response: We thank the reviewer for this recommendation. At this stage, we cannot modify our study protocol and recruitment is ongoing. Further, the recruitment would be much more difficult if we had to match GDM and non-GDM on several variables. However, we have added in the Strengths and Limitations section that the sample is relatively small and that there will be potential residual confounding.

Reviewer: 3

Reviewer Name: Delphine Mitanchez

This paper presents a study protocol. This study aims to evaluate cardiovascular risk factors in a cohort of newborn of mothers with gestational diabetes compared to newborns of mothers without diabetes. The evaluation is based primarily on cardiac and vascular ultrasound data at birth. The theme of programming is an important topic. There is little data on the specific subject of cardiovascular risk factors identified from the perinatal period. The protocol is well described but it needs minor revisions.

The introduction of the paper is too long. Two paragraphs could be deleted p 5, text between "Several epidemiological...prepubertal offspring".

Response: As suggested, we have shortened the introduction section and deleted these two paragraphs.

In the method part, what the authors call neonatal cardiovascular risk factors (secondary outcomes) are in fact biological factors (metabolic and inflammatory). There is currently no evidence of an association between the values of these factors in cord blood and cardiovascular risk factors.

Response: We have added that these factors could be “potential” cardiovascular risk factors.

Offspring assessment: Could the authors give precision about miRNA they aim to study in cord blood. Moreover, details should be provided on the growth parameters collected at birth and the reference curves used.

Response: Regarding miRNA, we have added the following sentence: “We will explore the profile of circulating non-coding RNAs in offspring of women with GDM and without GDM to assess the involvement of epigenetic mechanisms in cardiovascular alterations”. Regarding growth parameters collected at birth and the reference curves, we have added the following sentence: “The height and weight of the neonate will be measured and the growth reference charts of the Swiss Society of Paediatrics will be used [Braegger 2011, De Onis 2004]” [with these references: Braegger C, Jenni O, Konrad D, et al. Nouvelles courbes de croissance pour la Suisse. *Paediatr* 2011;22:9-11 and de Onis M, Garza C, Victora CG, et al. The WHO Multicentre Growth Reference Study. *Food Nutr Bull* 2004;25(1 Suppl):3-84].

Fetal echocardiography: which reference curves will be used to determine the z-score of the cardiac ultrasound measurements.

Response: We have specified in the methods section the reference curves that will be used for several measurements and added the following references:

- 1) for cardiovascular measurement: Gagnon C, Bigras JL, Fouron JC, et al. Reference Values and Z Scores for Pulsed-Wave Doppler and M-Mode Measurements in Fetal Echocardiography. *J Am Soc Echocardiogr* 2016;29(5):448-60.
- 2) for liver volume: Boito SM, Struijk PC, Ursem NT, et al. Assessment of fetal liver volume and umbilical venous volume flow in pregnancies complicated by insulin-dependent diabetes mellitus. *Bjog* 2003;110(11):1007-13
- 3) for umbilical artery pulsatility index: Ebbing C, Rasmussen S, Kiserud T. Middle cerebral artery blood flow velocities and pulsatility index and the cerebroplacental pulsatility ratio: longitudinal reference ranges and terms for serial measurements. *Ultrasound Obstet Gynecol* 2007;30(3):287-96.
- 4) for uterine artery pulsatility index: Gomez O, Figueras F, Fernandez S, et al. Reference ranges for uterine artery mean pulsatility index at 11-41 weeks of gestation. *Ultrasound Obstet Gynecol* 2008;32(2):128-32.

The authors mention in the statistical analysis the consideration of maternal treatment for diabetes. How will they take into account the quality of maternal diabetes control that can influence fetal development?

Response: Stratified analyses by the level of diabetes control will be conducted. Furthermore, multivariate analyses with adjustment for diabetes treatment will be done.

Reviewer: 4

Reviewer Name: Tine Dalsgaard Clausen

This is a protocol describing a study, which started inclusion in September 2016. It is of course an editorial decision to publish research protocols, however I would personally only very seldom be a reader of protocols as presented here. The scope to evaluate cardiovascular risk in GDM offspring is very relevant. It is measuring findings from fetal echo as well as echo in newborn+carotis intima measures as proxy for later cardiovascular risk in offspring of mothers with GDM compared to non-GDM mothers. The authors refer to a single reference claiming that the methods (when used in children and adolescents) are valid predictors of cardio-vascular disease in later life. However I am

not convinced that the methods have been validated when used in fetuses and newborn. A long-term follow up would therefore be extremely important to evaluate whether early changes is associated with measures later in childhood – newborn cardiomyopathy is often reversible. However – a long-term follow up from 2x100 offspring will probably be underpowered due to lost-to follow-up, and it has not been funded yet.

Response: We essentially agree with the reviewers. However, we have not yet funded this long-term follow-up. Furthermore, for such a long-term follow-up, the study protocol (including statistical power calculation) and the outcomes will have to be defined accounting for the relatively small sample. At this stage, we prefer not to give further details. We have ensured that participants agreed for a potential longer follow-up.

Furthermore the authors claim that the study has several advantages compared to other follow-up studies –I agree that the prospective design is admirable, however it does not at all eradicate risk of confounding, which the authors claims. Furthermore inconsistent results from previous studies cannot be referred to as “methodological flaws”.

Response: We have now acknowledged the limitations and risk of bias in our study. Further, we have deleted the term “inconsistent results”.

I miss data on intrauterine growth (UL sound measures).

Response: We have added the following reference for the intrauterine growth measurements: Papageorgiou AT, Ohuma EO, Altman DG, et al. International standards for fetal growth based on serial ultrasound measurements: the Fetal Growth Longitudinal Study of the INTERGROWTH-21st Project. Lancet 2014;384(9946):869-79. Furthermore, in the Foetal echography section, we have added the following sentences: “Observed foetal weight (FW) will be calculated using the Hadlock formula based on the head circumference, biparietal diameter, abdominal circumference and femur length [Hadlock 1985]. The gestational age-adjusted expected FW will be calculated [Salomon 2007] and the discordance with observed FW will be determined as (observed FW-expected FW)/expected FW. This discordance will be used to quantify appropriateness of foetal growth”.

Finally, as I read the protocol it is linked to another protocol – a RCT which aim to evaluate the effect of intensified intervention in GDM pregnancies. This is a relevant scope; however inclusion of 100 GDM pregnancies would be extremely underpowered to evaluate any effect . So - the study will in theory be able to evaluate early cardio-vascular risk indicators, however it will not have data to evaluate potential effects of a more tight glycemic on the cardiovascular risk in the offspring.

Response: There is a coordination between our study and the RCT, especially to help recruitment and, when relevant, to share resources (laboratory, human resources).

The RCT is ongoing and registered online (<https://clinicaltrials.gov/ct2/show/NCT02890693>), and the study protocol will be soon available. The primary outcomes are maternal outcomes. We are not responsible for the design of this trial and have no comments to make on its statistical power. Besides the close coordination between the two studies, this trial has no impact on our cohort study.

VERSION 2 – REVIEW

REVIEWER	Kelli Ryckman University of Iowa, Iowa City, IA, USA
REVIEW RETURNED	03-Aug-2017

GENERAL COMMENTS	All of my previous comments have been addressed.
--

REVIEWER	Tine Dalsgaard Clausen Department of Gynecology and Obstetrics Nordsjællands Hospital Hillerød Denmark
REVIEW RETURNED	15-Aug-2017

GENERAL COMMENTS	none
------